# COVID-19 incidence and mortality in non-dialysis chronic kidney disease patients

**Dino Gibertoni**[1☯]*, **Chiara Reno**[1☯], **Paola Rucci**[1], **Maria Pia Fantini**[1], **Andrea Buscaroli**[2], **Giovanni Mosconi**[3,4], **Angelo Rigotti**[5], **Antonio Giudicissi**[4], **Emanuele Mambelli**[5], **Matteo Righini**[2], **Loretta Zambianchi**[3], **Antonio Santoro**[6], **Francesca Bravi**[7], **Mattia Altini**[7]

**1** Department of Biomedical and Neuromotor Sciences, University of Bologna, Bologna, Italy, **2** Unit of Nephrology and Dialysis, "Santa Maria delle Croci" Hospital, Ravenna, Italy, **3** Unit of Nephrology and Dialysis, "Morgagni-Pierantoni" Hospital, Forlì, Italy, **4** Unit of Nephrology and Dialysis, "M. Bufalini" Hospital, Cesena, Italy, **5** Unit of Nephrology and Dialysis, "Infermi" Hospital, Rimini, Italy, **6** Specialty School of Nephrology, University of Bologna, Bologna, Italy, **7** Local Healthcare Authority of Romagna (AUSL Romagna), Ravenna, Italy

☯ These authors contributed equally to this work.
\* dino.gibertoni2@unibo.it

## Abstract

Many studies reported a higher risk of COVID-19 disease among patients on dialysis or with kidney transplantation, and the poor outcome of COVID-19 in these patients. Patients in conservative management for chronic kidney disease (CKD) have received attention only recently, therefore less is known about how COVID-19 affects this population. The aim of this study was to provide evidence on COVID-19 incidence and mortality in CKD patients followed up in an integrated healthcare program and in the population living in the same catchment area. The study population included CKD patients recruited in the Emilia-Romagna Prevention of Progressive Renal Insufficiency (PIRP) project, followed up in the 4 nephrology units (Ravenna, Forlì, Cesena and Rimini) of the Romagna Local Health Authority (Italy) and alive at 1.01.2020. We estimated the incidence of COVID-19, its related mortality and the excess mortality within this PIRP cohort as of 31.07.2020. COVID-19 incidence in CKD patients was 4.09% (193/4,716 patients), while in the general population it was 0.46% (5,195/1,125,574). The crude mortality rate among CKD patients with COVID-19 was 44.6% (86/193), compared to 4.7% (215/4,523) in CKD patients without COVID-19. The excess mortality of March-April 2020 was +69.8% than the average mortality of March-April 2015–19 in the PIRP cohort. In a cohort mostly including regularly followed up CKD patients, the incidence of COVID-19 among CKD patients was strongly related to the spread of the infection in the community, while its lethality is associated with the underlying kidney condition and comorbidities. COVID-19 related mortality was about ten times higher than that of CKD patients without COVID. For this reason, it is urgent to offer a direct protection to CKD patients by prioritizing their vaccination.

emilia-romagna.it/) and, although they are anonymized, datasets are not publicly available due to the current regulation on privacy. The description of the administrative databases is available from the website https://salute.regione. emilia-romagna.it/siseps/sanita/asa/ documentazione. Other researchers can obtain access to the data through a formal request based on a research project to the Romagna Local Health Authority.

**Funding:** The author(s) received no specific funding for this work.

**Competing interests:** The authors have declared that no competing interests exist.

## Introduction

The outbreak of a new epidemic carries along the urgent need to understand how the different segments of the population are affected in order to develop appropriate public health strategies to protect vulnerable subgroups and to support clinical management and decision-making. The emergence of a novel coronavirus in Wuhan, China, in December 2019, isolated in early January 2020 and later on in February named severe acute respiratory syndrome coronavirus 2 (SARS-CoV-2) causing coronavirus disease 2019 (COVID-19) [1] prompted global investigations on the effects and clinical manifestations of the virus and, on the other side, on the possible risk factors for a severe disease. The virus has spread at an impressive and alarming speed around the world and on March 11, 2020, the World Health Organization (WHO) declared COVID-19 a pandemic [2]. Italy has been one of the first European countries to be affected by the pandemic and during the first wave Northern regions were markedly involved. However, even within the same region, a different geographical pattern of virus spread occurred depending on the distance from disease hotspots, thus leading areas of the same territory to suffer the effects of the pandemic with different timings [3].

Evidence accumulating over time showed that, although SARS-CoV-2 infection primarily causes respiratory illness with highly variable clinical manifestations, other organs may be damaged by the virus, the kidney being one of the main site of complications [4]. Patients requiring dialysis and kidney transplant recipients have been first identified as a subgroup at higher risk for poor outcomes, and often present atypical clinical features that constitute an additional challenge [5]. Only recently chronic kidney disease (CKD) has been demonstrated to be a key risk factor for COVID-19 mortality as well, with a clear association between the level of dysfunction and mortality rate [6–8]. Specifically, in the recently published OpenSA-FELY project, based on 17 million patients [6, 7], dialysis (adjusted hazard ratio (aHR) = 3.69), organ transplant (aHR = 3.53) and stage of CKD (aHR = 2.52 for patients with eGFR <30 mL/ min/1.73 m$^2$) were three of the four comorbidities associated with the highest mortality risk from COVID-19. The risk associated with low eGFR was higher than the risk associated with diabetes mellitus (aHR range 1.31–1.95, depending upon the level of glycaemic control) or chronic heart disease (aHR = 1.17). In another recent publication of the Global Burden of Disease collaboration, CKD was reported as the most prevalent risk factor for severe COVID-19 [9]. The high prevalence of CKD among COVID-19 cases and their elevated risk of mortality calls for urgent action on this vulnerable population [10]. To date, evidence from the literature on this topic is based on hospitalized patients with COVID-19 and there is a lack of information on the incidence of COVID-19 from longitudinal studies conducted on CKD patients. To fill this gap, our study aimed at estimating the incidence and mortality related to COVID-19 in a large cohort of CKD patients enrolled in the Emilia-Romagna region project "Prevenzione insufficienza renale progressiva" (PIRP).

Our specific aims were: 1) to estimate the incidence of COVID-19 in CKD patients; 2) to identify predictors of the incidence of COVID-19 among CKD patients; 3) to compare the mortality of CKD patients with and without COVID-19; 4) to analyze predictors of COVID-19 mortality among CKD patients; 5) to estimate the excess mortality of CKD patients during the first wave of COVID-19 pandemic compared to the equivalent period in 2015–19; 6) to compare in-hospital COVID-19 mortality between CKD and non-CKD patients.

## Materials and methods

The study was approved by the Romagna Ethics Committee on 9.10.2020 n° 7843/2020. Consent was not required because data were analyzed anonymously.

## Study population

The study population includes CKD patients enrolled in the PIRP project and residing in the Romagna Local Health Authority (AUSL Romagna) catchment area, that encompasses the provinces of Ravenna, Forlì-Cesena and Rimini.

PIRP is a project established in 2004 and funded by the Emilia-Romagna region that is devised to timely intercept and follow up people with chronic kidney disease with the aim to delay their progression and prevent kidney failure [11]. The project started in 2004 and to date includes more than 31,000 patients and 130,000 visits, being one of the largest European registries on CKD patients [12]. Patients followed up in PIRP are adult patients with CKD-EPI stage 3a to 5, or patients at an earlier CKD stage with albuminuria/proteinuria or abnormalities detected by renal imaging, with or without other comorbidities [11]. Patients who reach ESKD (dialysis or transplant) exit the project. Considering that the most recent estimate of CKD stage 3–5 prevalence in Italy was 2.89% [13], and that the prevalence of PIRP patients living in the AUSL Romagna area was 0.84%, the PIRP project gathered about 30% of CKD stage 3–5 patients. The database of the project includes information about ambulatory visits, laboratory data, drug prescriptions, and is linked with the official regional databases of dialysis and transplantation, the hospital discharge records (HDR) database, and the mortality registry ("Registro Mortalità"—ReM). Patients extracted for the present study were those alive on 01.01.2020 and in pre-dialysis CKD stages. They comprised both patients under periodical nephrological follow-up and those referred back to their general practitioner because they had a less severe degree of CKD, or to other specialists.

AUSL Romagna is a Local Health Authority of Northern Italy located in Emilia-Romagna region with a catchment area of 1,125,474 inhabitants as of January 1, 2020 and a population density of 220.6 per km$^2$. It comprises a high population density plain and seaside area, with several close and connected small- and middle-sized urban areas, and a low population density mountain area. The four nephrology units participating in the PIRP project and operating in the hospitals of Ravenna, Forlì, Cesena and Rimini were involved in the present study. They share common therapeutic guidelines for the treatment of CKD patients and have the same care pathway.

## Data sources

PIRP patients with COVID-19 disease were extracted from the HDR database or from the database of SARS CoV-2 positive patients confirmed by RT-PCR for the period between January 1, 2020 and July 31, 2020. The COVID-19 onset date was the admission date for those hospitalized and the referral date for those not hospitalized. Mortality in the PIRP cohort was obtained from the mortality registry (ReM) database, which includes deaths occurred both in- and out-of-hospitals and information on the causes of death.

Information on SARS-CoV-2 infection in the general population was obtained from the official figures provided by the Italian National Ministry of Health and released by the Civil Protection Department (CPD) on their website www.protezionecivile.gov.it. Residents as of January 1, 2020 were retrieved from the Italian National Institute of Statistics website (http://demo.istat.it). Mortality in the general population was retrieved from official data on COVID-19 released by AUSL Romagna up to October 25, 2020 [14].

Data on patients' comorbid conditions were retrieved from the HDR database. Specifically, cardiovascular diseases, COPD, tumors and liver disease were identified using the Elixhauser algorithm [15] by searching for all the diagnoses reported in HDR of 2017–19. Dementia was assessed by searching for a specific subset of diagnosis codes in the HDR database, and diabetes by combining data from the clinical history of the PIRP registry and the HDR database

(Table A in S1 Appendix). eGFR was estimated from patients' most recent creatinine values using the CKD-EPI equation [16] and CKD stages were defined according to KDIGO guideline.

## Statistical analysis

Clinical and demographic patients' characteristics were summarized using mean±SD or absolute and relative frequencies according to the type of variable. Multivariable logistic regression was used to identify PIRP patients' characteristics independently associated with COVID-19 disease.

COVID-19 related mortality rates were compared among CKD stages, sex and geographical area. Survival in subgroups of CKD patients was investigated using Kaplan-Meier survival analysis and log-rank tests. In this analysis, the period of COVID-19 onset was used as predictor to test whether the risk of mortality was higher in the first weeks of the pandemic.

The excess mortality in CKD patients was estimated for the year 2020. To this purpose, age and sex adjusted mortality rates were computed for the period January-July of each year between 2015 to 2020, using the age/sex distribution of the PIRP cohort on January 1, 2015. The excess mortality rate was then computed as the ratio of the adjusted mortality rate of 2020 to the mean of the adjusted mortality rates for the years 2015–19.

In-hospital COVID-19 mortality was compared between CKD patients and non-CKD patients by searching for all individuals with a hospital admission due to COVID-19 in the period 1 January– 31 July 2020 and performing a multivariable logistic regression. In this regression, age, sex, province and comorbidities assessed during hospital admissions of 2018–19 were included as potential confounders.

All analyses were carried out using Stata v.15.1, and the significance level was set to p<0.05.

## Results

### Characteristics of the study cohort

In the study area, 4716 patients followed up in the PIRP project were alive on January 1, 2020. Mean age was 76.2±11.6 years, males were 65.4%, eGFR was 38.1±15.8 ml/min/1.73m$^2$. 33.8% had diabetes, 21.0% cardio-vascular comorbidities and 5.1% COPD. The characteristics of CKD patients by CKD-EPI stage and by province are reported in the (Tables B-C in S1 Appendix).

### Incidence of COVID-19

As of July 31, 2020, 193/4716 CKD patients had COVID-19 disease, with a 4.09% incidence. Among these 184/193 (95.4%) were hospitalized. In the general population of AUSL Romagna, the incidence of COVID-19 cases was 0.46%. In the province of Rimini there was the highest incidence of COVID-19, both in the general population (0.66%) and among CKD patients (5.35%), and in Ravenna the lowest (0.29% and 2.58% respectively).

CKD patients affected by COVID-19 were on average older than those unaffected (80.8 vs. 76.0 years), had a poorer kidney function (eGFR = 32.5 vs. 38.3 mL/min/1.73m$^2$) and a higher prevalence of cardiovascular comorbidities, chronic obstructive pulmonary disease (COPD) and tumors (Table 1). Multivariable logistic regression confirmed that patients with CKD-EPI stage 4 (OR = 1.562), older age (OR = 1.033), cardiovascular comorbidities (OR = 1.822), COPD (OR = 1.906) and tumors (OR = 1.631) were more likely to have COVID-19. After adjustment for covariates, patients living in the Ravenna province (OR = 0.491) and those living in the Forlì-Cesena province (OR = 0.662) were less likely to be affected by COVID-19 disease than those living in Rimini.

**Table 1. Characteristics of CKD patients with and without COVID-19 disease, and predictors of COVID-19.**

|  | With COVID-19 (n = 193) | Without COVID-19 (n = 4523) | OR (95% CI) | p-value |
|---|---|---|---|---|
| Age at January 1, 2020, mean±SD | 80.8±8.9 | 76.0±11.7 | 1.033 (1.014–1.053) | 0.001 |
| Males, n(%) | 123(63.7) | 2963(65.5) | 1.205 (0.872–1.666) | 0.258 |
| Immigrants, n(%) | 6(3.1) | 170(3.8) | 1.301 (0.508–3.327) | 0.583 |
| Province, n(%) |  |  |  |  |
| Ravenna | 22(11.4) | 830(18.3) | 0.491 (0.303–0.796) | 0.004 |
| Forlì-Cesena | 62(32.1) | 1765(39.0) | 0.662 (0.472–0.928) | 0.017 |
| Rimini | 109(56.5) | 1928(42.6) | Ref. |  |
| BMI (kg/m$^2$), mean±SD | 28.4±5.1 | 28.0±4.8 |  |  |
| <25 | 51(27.4) | 1240(27.8) | Ref. |  |
| 25–29.99 | 69(37.1) | 1912(42.8) | 0.874 (0.598–1.279) | 0.489 |
| 30–34.99 | 47(25.3) | 969(21.7) | 1.197 (0.783–1.829) | 0.406 |
| ≥35 | 19(10.2) | 343(7.7) | 1.424 (0.803–2.526) | 0.226 |
| eGFR (mL/min/1.73m$^2$), mean±SD | 32.5±12.1 | 38.3±15.9 |  |  |
| CKD-EPI stage 1–2 | 4(2.1) | 336(7.4) | 0.575 (0.203–1.628) | 0.298 |
| CKD-EPI stage 3a | 25(12.9) | 989(21.9) | 0.728 (0.447–1.187) | 0.204 |
| CKD-EPI stage 3b | 73(37.8) | 1772(39.2) | Ref. |  |
| CKD-EPI stage 4 | 84(43.5) | 1256(27.8) | 1.562 (1.116–2.186) | 0.009 |
| CKD-EPI stage 5 | 7(3.6) | 169(3.7) | 1.090 (0.485–2.449) | 0.834 |
| Primary kidney disease, n(%) |  |  |  |  |
| Hypertensive nephropathy | 150(77.7) | 3124(69.1) |  |  |
| Diabetic nephropathy | 17(8.8) | 415(9.2) |  |  |
| Polycystic kidney | 3(1.6) | 95(2.1) |  |  |
| Pyelonephritis | 2(1.0) | 227(5.0) |  |  |
| Glomerulonephritis | 4(2.1) | 156(3.4) |  |  |
| Single kidney | 7(3.6) | 219(4.8) |  |  |
| Unknown nephropathy | 8(4.2) | 225(5.0) |  |  |
| Rare nephropathies | 2(1.0) | 62(1.4) |  |  |
| Diabetes, n(%) | 69(35.7) | 1500(33.2) | 1.033 (0.748–1.427) | 0.845 |
| Cardiovascular comorbidities, n(%) | 77(39.9) | 911(20.2) | 1.822 (1.311–2.533) | <0.001 |
| COPD, n(%) | 26(13.5) | 213(4.7) | 1.906 (1.182–3.072) | 0.008 |
| Tumors, n(%) | 21(10.9) | 300(6.6) | 1.631 (1.002–2.655) | 0.049 |
| Liver disease, n(%) | 2(1.0) | 41(0.9) | - | - |
| Dementia, n(%) | 10(5.2) | 96(2.1) | 1.650 (0.827–3.292) | 0.155 |

## Mortality

**Predictors of COVID-19 mortality in CKD patients.** Overall, 301/4716 (6.4%) CKD patients died up to July 31, 2020 (Table 2). The crude mortality rate in CKD patients with COVID-19 (86/193, 44.6%) was almost tenfold than in those without COVID-19 (215/4523, 4.7%). COVID-19 mortality in CKD patients was higher among those in CKD-EPI stage 4 (54.8%) and was consistently higher than in patients without COVID-19 in all CKD stages. The CMR of CKD patients with COVID-19 did not show remarkable differences between sex and among provinces.

Factors significantly associated with higher COVID-19 mortality in CKD patients in univariate survival analysis (Fig 1 and Table D in S1 Appendix) were older age, most severe stages of CKD, dementia and SARS-CoV-2 infection in the early phase of the pandemic. Notably, the large majority of patients (71.4%) who were infected between March 8–21, 2020 died.

**Table 2. Mortality in CKD patients with and without COVID-19 disease.**

| | All CKD patients | | No COVID-19 disease | | COVID-19 disease | |
|---|---|---|---|---|---|---|
| | Number of deaths | Crude mortality rate | Number of deaths | Crude mortality rate | Number of deaths | Crude mortality rate |
| Overall population | 301/4716 | 6.38% | 215/4523 | 4.75% | 86/193 | 44.56% |
| CKD-EPI stage | | | | | | |
| 1–2 | 3/340 | 0.88% | 2/336 | 0.60% | 1/4 | 25.00% |
| 3a | 30/1014 | 2.96% | 23/989 | 2.33% | 7/25 | 28.00% |
| 3b | 109/1845 | 5.91% | 80/1772 | 4.51% | 29/73 | 39.73% |
| 4 | 138/1340 | 10.30% | 92/1256 | 7.32% | 46/84 | 54.76% |
| 5 | 21/176 | 11.93% | 18/169 | 10.65% | 3/7 | 42.86% |
| Males | 175/3086 | 5.67% | 121/2963 | 4.08% | 54/123 | 43.90% |
| Females | 126/1630 | 7.73% | 94/1560 | 6.03% | 32/70 | 45.71% |
| Province | | | | | | |
| Ravenna | 53/852 | 6.22% | 43/830 | 5.18% | 10/22 | 45.45% |
| Forlì-Cesena | 121/1827 | 6.62% | 92/1765 | 5.21% | 29/62 | 46.77% |
| Rimini | 127/2037 | 6.23% | 80/1928 | 4.15% | 47/109 | 43.12% |

The cause of death was reported in the mortality registry for 84/86 CKD patients and was COVID-19 in 30/84 (35.7%) cases; in 7/84 (8.3%) cases pneumonia and respiratory diseases, and in 4/84 (4.8%) cases infectious diseases were reported. However, 61/86 (70.9%) CKD patients died during the hospitalization for COVID-19 disease. Acute kidney injury (AKI) was developed by 21/193 (10.9%) patients, of whom 11/21 (52.4%) died. Only 4/193 (2.1%) underwent hemodialysis treatment during their COVID hospitalization, and 2 of them had a concurrent AKI.

To enable the comparison with hemodialysis patients, for whom the CMR in an Italian cohort is available as of April 23, 2020 [17], we computed the CMR of CKD patients at the same date. Results indicate that the CMR was slightly lower in CKD patients than in hemodialysis patients (37.8% vs. 41.5%).

**Excess mortality in CKD patients.** In the overall PIRP cohort, mortality in January-July of 2020 increased by +17.7% with respect to the average mortality observed in the same period in 2015–19 (Table 3). This increase corresponds to 77 more deaths more than expected. The excess mortality had a peak in the months of March and April, when 58 more deaths (+69.8%) were observed. In January-February, before the pandemic outbreak, a slight decrease of mortality was recorded, while from May to July an excess of 22 deaths (+10.1%) occurred.

**Predictors of in-hospital COVID-19 mortality in CKD patients vs. non-CKD patients.** The multivariable logistic regression model of in-hospital COVID-19 mortality revealed that, after adjusting for age, sex, area and comorbidities, CKD patients had a 43.8% higher risk of dying than the other hospitalized individuals (Table 4 and Table E in S1 Appendix).

## Discussion

In our cohort involving 4,716 CKD patients in non-dialysis stage, we found a 4.09% incidence of COVID-19 and a related mortality rate of 44.6%. Our results are consistent with evidence from the literature suggesting that CKD patients are on average older and more vulnerable to SARS-CoV-2 infection [18] and that CKD is an underlying condition that increases the risk of severe COVID-19 illness [19].

Older age, lower GFR, cardiovascular comorbidities and COPD were associated with a higher likelihood of SARS-CoV-2 infection, confirming previous findings [20, 21]. After

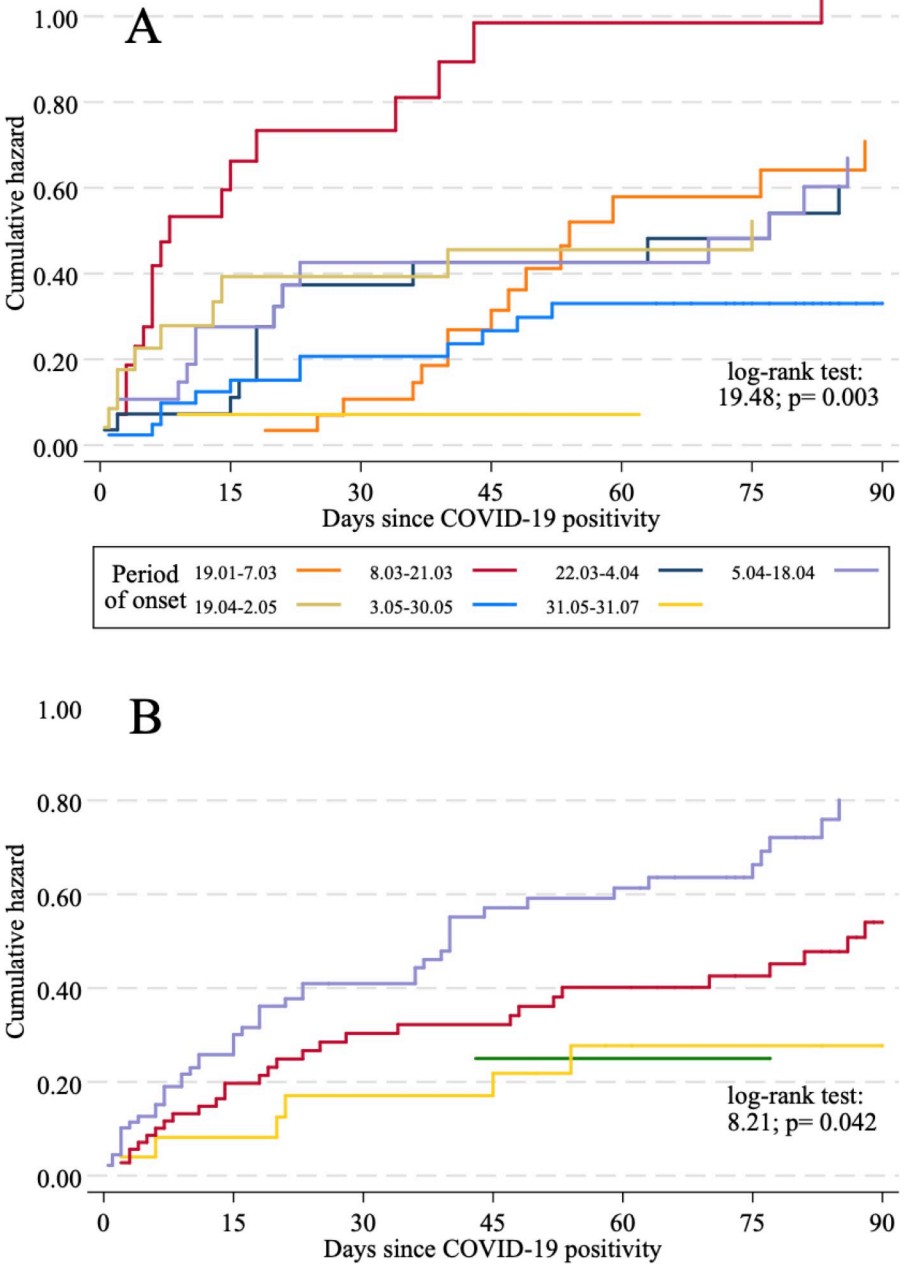

**Fig 1. Cumulative hazards of COVID-19 related mortality.** (A) By period of onset of the disease. (B) By CKD stage.

adjusting for these conditions, the geographical area remained an independent risk factor, underscoring that the incidence of COVID-19 in CKD patients reflects that of the general population of the same area. Specifically, the Rimini province was exposed to an early intense epidemic flow originating from the neighboring hotspot area of Pesaro (Marche region), which happened before the national lockdown enforced on March 9, 2020 by the Italian Government [22] when healthcare services had to face an exceedingly large number of people referring to

**Table 3. Excess mortality of CKD patients in 2020 with respect to the average mortality of 2015–19.**

| Period | Number of deaths | | | | Adjusted mortality rates | | Excess mortality (%) | Excess mortality (n) |
|---|---|---|---|---|---|---|---|---|
| | Average 2015–19 | 2020 | 2020 COVID disease | 2020 no COVID disease- | Average 2015–19 | 2020 | | |
| January-February | 82 | 79 | 0 | 79 | 1.82% | 1.53% | -15.9% | -3 |
| March-April | 65 | 123 | 51 | 72 | 1.44% | 2.44% | +69.8% | 58 |
| May-July | 80 | 102 | 35 | 67 | 1.80% | 1.98% | +10.1% | 22 |
| *January-July* | *227* | *304* | *86* | *218* | *5.06%* | *5.95%* | *+17.7%* | *77* |

hospitals and emergency wards. The incidence of COVID-19 in CKD patients differed among the provinces examined in our study, although access of CKD patients to ambulatory facilities was similar and based on a standardized care pathway [11] even during the pandemic period.

It must be underlined that our data relate to the first phase of the pandemic, from February to July 2020, when no screening program for vulnerable categories was enforced to identify and trace COVID-19. As CKD patients are mostly old and multimorbid, they are particularly at risk to have symptomatic COVID-19 disease, therefore it is likely that they had been relatively more tested than the general population in that phase of the pandemic. Moreover, patients of our cohort are routinely followed up in a prevention program and are aware of their frail condition, which could have enhanced their propensity to be timely checked for COVID-19.

The high risk of COVID-19 mortality in CKD patients was confirmed by several comparisons carried out in this study. The crude mortality rate of 44.6% was of similar magnitude to that observed in hemodialysis patients treated in the dialysis units of the Emilia-Romagna region [17] and higher than the 25–30% range reported for people older than 70 years in Italy [23, 24] and the 32.9% reported for CKD patients in the German Lean European Open Survey [25].

Lastly, in-hospital mortality was significantly higher than that of non-CKD patients hospitalized for COVID-19 in the same area and period. The CMR was very similar across geographical areas, regardless of the different incidence rates. In fact, once COVID-19 disease is established, CKD patients have high mortality rates because the disease severely affects their already impaired kidney function [4]. These figures consistently underscore the high

**Table 4. Predictors of in-hospital COVID-19 mortality.**

| | Odds Ratio | 95% CI | p-value |
|---|---|---|---|
| CKD (enrolled in PIRP) | 1.438 | 1.046–1.978 | 0.025 |
| Age, years | 1.069 | 1.062–1.076 | 0.000 |
| Males | 1.414 | 1.226–1.631 | 0.000 |
| COPD | 1.038 | 0.785–1.371 | 0.794 |
| Liver disease | 1.382 | 0.763–2.505 | 0.286 |
| Dementia | 1.792 | 1.318–2.437 | 0.000 |
| Cardiovascular comorb. | 1.282 | 1.054–1.559 | 0.013 |
| Diabetes | 1.198 | 0.915–1.569 | 0.188 |
| Tumors | 1.685 | 1.317–2.154 | 0.000 |
| Province | | | |
| Ravenna | 0.870 | 0.730–1.037 | 0.120 |
| Forlì-Cesena | 1.203 | 1.023–1.416 | 0.026 |
| Rimini | Ref. | | |

vulnerability of CKD patients to COVID-19, related to multiple factors [26, 27], and the need to protect this segment of the population.

Another way to look at the COVID-19 death toll on CKD patients is to examine the excess mortality. This measure is free of the potential biases due to incorrect or lacking attribution of the COVID-19 disease to subjects, and as such it allows to estimate the impact of COVID-19 on the overall mortality. CKD patients experienced in the period February 24 –April 30, 2020 an excess mortality of +69.8%, while in the same interval the excess mortality rates of the general population of AUSL Romagna aged 75 years or more were lower [3]. As of July 31, the total excess deaths among CKD patients were 77, and the number of deaths among CKD patients with COVID was 86, being COVID-19 almost the only determinant of excess mortality. CKD patients without COVID-19 had a slightly lower mortality than expected, which may be a positive side effect of the prevention measures enforced to avoid COVID-19 contagion, combined with the effectiveness of their disease management in the PIRP program.

Our study has several limitations. The study population refers to a specific area of Northern Italy, therefore our findings cannot be generalized to other regions or countries where the pandemic had different spread or other containment measures or treatment options were adopted. The number of COVID-19 cases and deaths is relatively small, which prevented us from conducting multivariable analyses on the predictors of mortality. Information on treatment during hospitalization is not available in hospital discharge records. Our results are based on CKD patients included in a prevention and treatment program which approximately covers 30% of CKD patients in AUSL Romagna catchment area, thus they cannot be generalized to all CKD patients.

However, our focus on a cohort of patients with CKD included in a registry and the availability of high-quality data individually linked to administrative databases represent a strength of our study because most of the available evidence on the effect of COVID-19 in CKD patients is derived from hospitalized patients.

In conclusion, we found that CKD patients in non-dialytic stage exhibit a vulnerability to COVID-19 disease comparable to that of patient on kidney replacement therapy. Despite the presence of effective clinical pathways for CKD patients as the PIRP program, the incidence of COVID-19 is strongly related to the spread of the infection in the community. While efforts are made to contain the outbreak with several non-pharmaceutical interventions and a massive vaccination campaign aimed to achieve herd immunity in the general population (indirect protection), it is urgent to give direct protection to CKD patients through their prompt vaccination. Therefore, it is of crucial importance to carefully develop strategies that target this vulnerable segment of the population at higher risk of severe or fatal outcome both in the current and future vaccination campaigns, taking into account the duration of protection, the emergence of new variants of SARS-CoV-2 and their transmission and lethality potential. In addition, consistent with recent action points outlined by ERA-EDTA council [10], we also advocate the inclusion of CKD patients in clinical trials testing the efficacy of drugs and vaccines to prevent severe COVID-19.

## Supporting information

**S1 Appendix. Supporting Information, including Tables A, B, C, D and E.**
(DOCX)

## Author Contributions

**Conceptualization:** Dino Gibertoni, Paola Rucci, Maria Pia Fantini.

**Data curation:** Dino Gibertoni.

**Formal analysis:** Dino Gibertoni.

**Investigation:** Chiara Reno.

**Software:** Dino Gibertoni.

**Supervision:** Maria Pia Fantini.

**Writing – original draft:** Dino Gibertoni, Chiara Reno.

**Writing – review & editing:** Dino Gibertoni, Chiara Reno, Paola Rucci, Maria Pia Fantini, Andrea Buscaroli, Giovanni Mosconi, Angelo Rigotti, Antonio Giudicissi, Emanuele Mambelli, Matteo Righini, Loretta Zambianchi, Antonio Santoro, Francesca Bravi, Mattia Altini.

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
