## [Decision Letter · Decision Letter 0]

21 Apr 2021

PONE-D-21-06894

COVID-19 incidence and mortality in pre-dialysis chronic kidney disease patients

PLOS ONE

Dear Dr. Gibertoni,

Thank you for submitting your manuscript to PLOS ONE. After careful consideration, we feel that it has merit but does not fully meet PLOS ONE’s publication criteria as it currently stands. Therefore, we invite you to submit a revised version of the manuscript that addresses the points raised during the review process.

**The manuscript focuses on a topic of potential interest. The study, however, presents several major shortcomings that need to be addressed. To mention some of them, i) concern about the fact that the comparison made between groups seem inappropriate, ii) consider as more appropriate comparison, sick elderly cohort without CKD, iii) unclear whether the PRIP cohort was tested routinely, or more frequently than the general population**; **iv) concern about the fact that if the cause of death was COVID-19 in only 37.5% of these cases, this may undermine the author main point, i.e. that these patients had excess death due to COVID-19 infection; v) it is difficult to say that CKD was responsible for infection or death in the patients with COVID-19; vi) unclear what is the adjusted r^2^ for the model of mortality attributed CKD; vii) need to add BMI, ethnicity, social deprivation, liver disease and HIV infection, known risk factors for death in COVID, and assess how do they interact with the model; viii) concern about the fact that the mortality is really high; ix) need to define better the study population; x) unclear whether the admissions to hospital were for COVID-19, or unrelated admissions with incidental tests; xi) concern about the fact that there was no specific research question formulated related to differences between provinces; xii) concern about the fact that the results are described separately for the three provinces included in the region, xiii) concern about the conclusion that the incidence of COVID-19 was higher in CKD patients than in the general population, not justified since there was no universal screening for COVID-19; xiv) need to provide in Table A the distribution over CKD stages; xv) need to explain in the section on mortality the difference between the terms crude mortality rate and case fatality rate.**

We look forward to receiving your revised manuscript.

Kind regards,

Giuseppe Remuzzi

Academic Editor

PLOS ONE

Journal Requirements:

3. Please include a copy of Table 6 which you refer to in your text on page 11.

<h3>** **</h3>

4. We note you have included a table to which you do not refer in the text of your manuscript. Please ensure that you refer to Table 5 in your text; if accepted, production will need this reference to link the reader to the Table.

Reviewers' comments:

Reviewer's Responses to Questions

**Comments to the Author**

1. Is the manuscript technically sound, and do the data support the conclusions?

Reviewer #1: Yes

Reviewer #2: No

2. Has the statistical analysis been performed appropriately and rigorously? 

Reviewer #1: Yes

Reviewer #2: No

3. Have the authors made all data underlying the findings in their manuscript fully available?

Reviewer #1: Yes

Reviewer #2: Yes

4. Is the manuscript presented in an intelligible fashion and written in standard English?

Reviewer #1: Yes

Reviewer #2: Yes

5. Review Comments to the Author

Reviewer #1: This study describes the incidence and outcome of COVID-19 in a cohort of CKD patients as compared to the general population living in the same region. The main finding is that both the incidence of COVID-19 and the mortality rate were around 10 times higher in CKD patients. This corresponds with the data of the OpenSAFELY study, where CKD was found to be major risk factor for COVID-19 related mortality.

Comments:

1. The results are described separately for the three provinces included in the region. Although there are some differences between the provinces, I doubt whether this affects the analysis for the main study outcomes. There was no specific research question formulated related to differences between provinces. I therefore belief that this aspect gets too much attention in the manuscript. For the primary analyses it suffices to use the entire population. Subsequently, it can be checked whether the main conclusions would change when provinces are analysed separately.

2. The conclusion that the incidence of COVID-19 was higher in CKD patients than in the general population might not be justified since there was no universal screening for COVID-19. It is imaginable that CKD patients have more severe symptoms of COVID-19 and that the diagnosis is more often made than in the general population in which there might be a higher proportion of ‘subclinical’ COVID-19 cases. This issue should be discussed in more detail.

3. In Table A, the main diagnosis in CKD patients is ‘hypertensive nephropathy’. Probably this diagnosis was used for patients who had CKD and were hypertensive. However, since most kidney diseases are accompanied by hypertension, this diagnosis should be reserved for patients in whom no other kidney disease is present and kidney biopsy shows characteristic vascular lesions. In fact, it appears that the underlying cause of CKD was unknown in the majority of patients.

4. Table A should provide the distribution over CKD stages

5. In Table A, the abbreviation BPCO should be explained. (Should be COPD?)

6. From Table 1 it is clear that the incidence of COVID-19 is much higher in CKD patients than in the general population. It is confusing that on P8, L154 it is stated that the proportion of cases in the CKD population was lower than in the general population.

7. In the section on mortality, the authors use the terms crude mortality rate and case fatality rate. They should explanin the difference, if any, or use the same term throughout the manuscript.

8. While the CKD stage is an important predictor of mortality, this is not mentioned in the text on p11.

9. On page 11, line 202 there is referral to Table 6. This should be Table 5.

Reviewer #2: Dr Gibertoni and colleagues present a descriptive study of COVID-19 infection in an Italian cohort, focusing on a subpopulation of patients with CKD and comparing disease incidence and mortality to the general population and to those within the cohort with a + COVID test to those without any test. These are interesting data and the linkage of multiple datasets is commendable. The section on excess mortality is the strongest, and is enlightening. However I am concerned that the comparator groups are inappropriate throughout and the high mortality rate and hospitalisations may be misattributed to CKD as a result, leading to incorrect conclusions have been drawn as a result. A number of the data presented seem to undermine the authors main point.

Major issues:

The biggest issue for me is that the comparisons being made between groups seem inappropriate.

Re: Incidence of COVID19

1. Its not ideal comparing these PRIP patients to the general population. These are clearly very elderly comorbid patients, whereas the general population will include healthy young people and potentially even children. A more appropriate comparison would be a sick elderly cohort without CKD. As a result the incidence data is of limited utility.

2. Were the PRIP cohort test routinely, or more frequently than the general population? Why? what was the rate of testing in each?

Re Cause of Death

1. If the cause of death was covid-19 in only 37.5% of these cases , does this not undermine the authors main point- that these patients had excess death due to COVID19 infection? Is this not compounded by the fact that 25% didn’t even die during the admission with COVID19?

Re: Mortality attributed to CKD

1. The patients with covid-19 were older, had more advanced renal disease, higher rates of CV disease, higher rates of COPD (Table 2 ). It is thus hard to say that CKD was responsible for infection or death here. I also suspect there was non-diagnosed covid19 in the “without covid19 group”.

2. What’s the adjusted r2 for this model? The OR for CV and COPD are rather large – the OR for eGFR is per unit, but very close to 1. How much a difference did this really make? ( what did using CKD stages reveal instead?)

3. BMI, ethnicity, social deprivation, liver disease and HIV infection are known risk factors for death in COVID- are these data available and how do they interact with the model? Does eGFR remain a predictor in this case? Given the advanced age of these patients, are there any data on dementia which is also a risk factor?

4. “The crude mortality rate 176 was almost tenfold in CKD patients with COVID-19 (86/193, 44.6%)…” Again these were predominantly hospitalised patients however? I think it’s a bit misleading to call these “CKD patients” when they have so many co-morbidities. The authors here would be more accurate saying there was a 10x death rate in older, sicker, more multimorbid patients in the PRIP cohort who were hospitalised than those who were not hospitalised. This is a very different sentiment and statement.

5. The official figures from CPD will presumably include patients with CKD as well as young adults, (children even?) etc? If the authors wish to demonstrate that CKD is a risk factor for death the control group should be similarly elderly multimorbid patients with just as much additional disease, but without CKD. Otherwise, their finding is that “elderly multimorbid patients” have a higher infection rate (?or testing rate) + mortality than an averaged population including young healthy patients. Is the general population age matched? Or matched for other co-morbidities?

6. At the risk of repeating myself, The mortality is really high –I do worry there’s a bias here to just capturing the patients sick enough to end up in hospital and thus skewing the fatality rate which is really a “fatality rate of covid19 admissions in elderly CKD patients. These are really sick old people – 87% CV disease, 38% COPD, 10% Cancer, 35% diabetes. I don’t think its accurate to attribute this very high mortality to CKD based on these data.

Re: PRIP cohort

7. The study population is not well defined here. It is not clear to what extent the study population has captured ALL CKD in the region. This is important because the authors label these as CKD patients and everyone else as “non CKD” – are these in fact referrals based on rapid progression or higher risk CKD? What % of the population with CKD are not in the PIRP cohort? Does the PRIP cohort include children and young people, or transplants on immunosupression? The authors mention the cohort includes those under follow up by nephrology and GP, but what is the breakdown of CKD stage? Does COVID mortality vary accordingly?

8. Were the admissions to hospital for COVID-19, or where these unrelated admission with incidental tests? What was the proportion? Admitted patients will have had multiple tests, so there may be a detection bias here.

9. The 95% hospitalisation rate seems extremely high – There must presumably be a major bias here towards the hospitalised patients as there was no routine testing in the community of the PRIP cohort so asymptomatic cases and mild disease that didn’t present for a test won’t have been captured. The COVID+ patients who were hospitalised will have had a test, but the COVID+ patients in the community may not. Unless there was routine testing, I don’t think these data have captured the full extent of COVID19+ in this population so I don’t think the 95% hospitalisation rate is valid.

Minor:

Line 183 the CFR is different to the rate at the start of the paragraph?

CKD stages mentioned in the statistical analysis section of methods but not in the paper

I appreciate there are geographical and cultural differences in practice, but Im not sure its reasonable to call these “pre-dialysis” patients - they are very sick patients who would not be offered RRT in many healthcare settings – these patients would be on conservative care pathways typically.

Figure 2a

I don’t think this is useful information and its very confusing to read – why is the date of onset of disease relevant? Why resulting in different death rates? Why are these seemingly arbitrary dates chosen?

6. PLOS authors have the option to publish the peer review history of their article (what does this mean?). If published, this will include your full peer review and any attached files.

Reviewer #1: **Yes: **L.B. Hilbrands

Reviewer #2: **Yes: **Eoin Daniel O'Sullivan

---

## [Author Response · Author response to Decision Letter 0]

8 Jun 2021

Reviewer #1: This study describes the incidence and outcome of COVID-19 in a cohort of CKD patients as compared to the general population living in the same region. The main finding is that both the incidence of COVID-19 and the mortality rate were around 10 times higher in CKD patients. This corresponds with the data of the OpenSAFELY study, where CKD was found to be major risk factor for COVID-19 related mortality.

Comments:

1. The results are described separately for the three provinces included in the region. Although there are some differences between the provinces, I doubt whether this affects the analysis for the main study outcomes. There was no specific research question formulated related to differences between provinces. I therefore belief that this aspect gets too much attention in the manuscript. For the primary analyses it suffices to use the entire population. Subsequently, it can be checked whether the main conclusions would change when provinces are analysed separately. 

We thank the reviewer for his suggestion. Indeed, the analyses by provinces took a great amount of space in the results without any background information provided in the Introduction. We have now added the rationale behind this analysis. In summary, this reflects our intention to highlight geographical differences in the incidence of COVID-19 that have occurred at a local level. 

2. The conclusion that the incidence of COVID-19 was higher in CKD patients than in the general population might not be justified since there was no universal screening for COVID-19. It is imaginable that CKD patients have more severe symptoms of COVID-19 and that the diagnosis is more often made than in the general population in which there might be a higher proportion of ‘subclinical’ COVID-19 cases. This issue should be discussed in more detail.

We acknowledged the reviewer’s suggestion to remove from the paper the part on the comparison of the incidence of COVID-19 between CKD patients and the general population. 

3. In Table A, the main diagnosis in CKD patients is ‘hypertensive nephropathy’. Probably this diagnosis was used for patients who had CKD and were hypertensive. However, since most kidney diseases are accompanied by hypertension, this diagnosis should be reserved for patients in whom no other kidney disease is present and kidney biopsy shows characteristic vascular lesions. In fact, it appears that the underlying cause of CKD was unknown in the majority of patients.

We agree with the reviewer that the “hypertensive nephropathy” category might be also related to patients with unknown nephropathy and hypertension. Thus, we removed this variable from the predictors used in the multivariable logistic regression model of COVID-19.

4. Table A should provide the distribution over CKD stages

We have now added information on CKD stages in the supplementary materials

5. In Table A, the abbreviation BPCO should be explained. (Should be COPD?)

We apologize for this mistake; indeed, it was COPD and we have now changed the acronym throughout the text. 

6. From Table 1 it is clear that the incidence of COVID-19 is much higher in CKD patients than in the general population. It is confusing that on P8, L154 it is stated that the proportion of cases in the CKD population was lower than in the general population.

That sentence was indeed not very clear. As it was related to Fig. 2 which has now been removed, we also deleted that sentence.

7. In the section on mortality, the authors use the terms crude mortality rate and case fatality rate. They should explan in the difference, if any, or use the same term throughout the manuscript.

We used the term crude mortality rate when referring to CKD patients without COVID-19 disease, and the term case fatality rate when referring to CKD patients with COVID-19 disease. However, to avoid possible confusion, in the current version of the manuscript we have always used the term crude mortality rate. 

8. While the CKD stage is an important predictor of mortality, this is not mentioned in the text on p11.

Thank you for pointing this out, we have now mentioned in text the relevance of CKD stage as predictor of mortality

9. On page 11, line 202 there is referral to Table 6. This should be Table 5.

We apologize for this mistake, that has been corrected.

 

Reviewer #2: Dr Gibertoni and colleagues present a descriptive study of COVID-19 infection in an Italian cohort, focusing on a subpopulation of patients with CKD and comparing disease incidence and mortality to the general population and to those within the cohort with a + COVID test to those without any test. These are interesting data and the linkage of multiple datasets is commendable. The section on excess mortality is the strongest, and is enlightening. However I am concerned that the comparator groups are inappropriate throughout and the high mortality rate and hospitalisations may be misattributed to CKD as a result, leading to incorrect conclusions have been drawn as a result. A number of the data presented seem to undermine the authors main point.

Major issues:

The biggest issue for me is that the comparisons being made between groups seem inappropriate.

Re: Incidence of COVID19

1. Its not ideal comparing these PRIP patients to the general population. These are clearly very elderly comorbid patients, whereas the general population will include healthy young people and potentially even children. A more appropriate comparison would be a sick elderly cohort without CKD. As a result the incidence data is of limited utility.

We agree with the reviewer that comparing the incidence of COVID-19 in PIRP patients to the general population is inappropriate. A cohort of elderly individuals without CKD to use as reference was not available. For this reason, also following rev. #1’s suggestion, we removed Table 1 and maintained only the general figure of incidence in the population, avoiding to highlight the comparison of incidences.

2. Were the PRIP cohort test routinely, or more frequently than the general population? Why? what was the rate of testing in each?

There is no screening program specifically addressed to the PIRP patients, thus we do not have information regarding their rate of testing. Supposedly, they are likely to have been tested more frequently because they are older, multi-morbid and more at risk of being symptomatic if infected. 

Re Cause of Death

1. If the cause of death was covid-19 in only 37.5% of these cases, does this not undermine the authors main point- that these patients had excess death due to COVID19 infection? Is this not compounded by the fact that 25% didn’t even die during the admission with COVID19?

We apologize for our lack of clarity regarding this point. First of all, it should be kept in mind that a relevant proportion of these deaths occurred when the cause of death coding was not yet updated to include COVID-19 disease. Thus, we have inevitably to deal with inaccuracy on this issue. By highlighting that 76.2% died during their hospitalization for COVID-19, we meant that this was the minimum proportion of CKD patients who, with the greatest likelihood, died of/for COVID-19.

Re: Mortality attributed to CKD

1. The patients with covid-19 were older, had more advanced renal disease, higher rates of CV disease, higher rates of COPD (Table 2 ). It is thus hard to say that CKD was responsible for infection or death here. I also suspect there was non-diagnosed covid19 in the “without covid19 group”. 

In Table 2 we only compared clinical characteristics of CKD patients with and without COVID-19, therefore we were not implying that CKD was the cause of the disease, because all these were CKD patients. We cannot rule out the presence of non-diagnosed COVID-19 in the without covid19 group, however, as it was previously pointed out, this should likely be a negligible proportion of patients, given their high-risk profile.

2.What’s the adjusted r2 for this model? The OR for CV and COPD are rather large – the OR for eGFR is per unit, but very close to 1. How much a difference did this really make? ( what did using CKD stages reveal instead?)

Please see the answer to point 3.

3. BMI, ethnicity, social deprivation, liver disease and HIV infection are known risk factors for death in COVID- are these data available and how do they interact with the model? Does eGFR remain a predictor in this case? Given the advanced age of these patients, are there any data on dementia which is also a risk factor?

We thank you for these suggestions and we think that he still refers to the predictive model of COVID-19 incidence described in Table 2. We have updated this model by adding BMI and dementia, and using CKD-EPI stages in place of eGFR. We did not include HIV (only 20 subjects among CKD patients were found HIV+, none of them with COVID-19), liver disease (only 1.0% of patients had liver disease), ethnicity (because only the Caucasian/Afro-american distinction was available) and social deprivation (data not available). We have also modified the identification of comorbidities. In the first version of the manuscript, we used the M-CDS algorithm which is based solely on drugs dispensations and, because of that, it overestimated the presence of COPD (as it includes also the dispensation of mucolytics and corticosteroid inhalers) and other comorbidities. We have now used the Elixhauser algorithm and searched for diagnoses in the HDR records of 2017-19. 

Results indicate that renal function is still a risk factor, with stage 4 having OR=1.562. CV and COPD are still risk factors as well, however, the OR of CV has decreased and the OR of COPD has increased. Tumors have now become a significant risk factor. Dementia was not much frequent and displayed an increased yet non-significant risk of COVID-19. The pseudo-R2 of this model is 0.068, however it must be kept in mind that the pseudo-R2 values are considerably lower than those of the *R*2 used in linear regression, and that pseudo-R2 values as low as 0.2-0.4 indicate excellent model fit. 

4. “The crude mortality rate 176 was almost tenfold in CKD patients with COVID-19 (86/193, 44.6%)…” Again these were predominantly hospitalised patients however? I think it’s a bit misleading to call these “CKD patients” when they have so many co-morbidities. The authors here would be more accurate saying there was a 10x death rate in older, sicker, more multimorbid patients in the PRIP cohort who were hospitalised than those who were not hospitalised. This is a very different sentiment and statement.

To address this and the following request by the reviewer we conducted an additional analysis based on all hospitalizations of January-July 2020 for COVID-19 disease. We then performed a multivariable logistic regression using mortality as the outcome to determine whether CKD patients in the PIRP cohort had a higher mortality risk than the rest of patients hospitalized for COVID-19, adjusting for age, sex, comorbidities and the province of residence. This model shows that PIRP patients had a 43.8% higher risk compared to non-PIRP patients.

As not all CKD patients participate in the PIRP project, we cannot rule out the possibility that other hospitalized were not identified as CKD. Keeping these limitations in mind, we think that this analysis may help elucidating the effect of CKD on COVID-19 related mortality.

5. The official figures from CPD will presumably include patients with CKD as well as young adults, (children even?) etc? If the authors wish to demonstrate that CKD is a risk factor for death the control group should be similarly elderly multimorbid patients with just as much additional disease, but without CKD. Otherwise, their finding is that “elderly multimorbid patients” have a higher infection rate (?or testing rate) + mortality than an averaged population including young healthy patients. Is the general population age matched? Or matched for other co-morbidities?

The reviewer is right in mentioning that the infection rate of CKD patients cannot be compared with that of the general population. We have now omitted this comparison from our results and reported the incidence in the general population at the beginning of the section “Incidence of COVID-19” only for descriptive purposes. 

6. At the risk of repeating myself, The mortality is really high –I do worry there’s a bias here to just capturing the patients sick enough to end up in hospital and thus skewing the fatality rate which is really a “fatality rate of covid19 admissions in elderly CKD patients. These are really sick old people – 87% CV disease, 38% COPD, 10% Cancer, 35% diabetes. I don’t think its accurate to attribute this very high mortality to CKD based on these data.

We understand that the in-hospital mortality of CKD patients is high. It is actually higher than that reported for CKD patients enrolled in other registries (32.9%, LEOSS Study Group, 2021). We have now cited LEOSS study in which 97.1% of CKD patients with PCR- or rapid-test confirmed infection were hospitalized, a figure consistent with our report (95.4%). 

As to the comorbidity, we have now used the standardized Elixhauser algorithm to identify comorbid conditions, as mentioned in the reply to point 3) and we found that the prevalence of comorbid conditions is lower. 

Re: PRIP cohort

7. The study population is not well defined here. It is not clear to what extent the study population has captured ALL CKD in the region. This is important because the authors label these as CKD patients and everyone else as “non CKD” – are these in fact referrals based on rapid progression or higher risk CKD? What % of the population with CKD are not in the PIRP cohort? Does the PRIP cohort include children and young people, or transplants on immunosupression? The authors mention the cohort includes those under follow up by nephrology and GP, but what is the breakdown of CKD stage? Does COVID mortality vary accordingly?

We thank the reviewer for this suggestion. In the ‘Study population’ section we have now described the inclusion criteria in the project, clarified that CKD patients enrolled in PIRP do not include all patients with CKD in their area, and provided an estimate of the population with CKD living in the AUSL Romagna area that are followed up in PIRP.

In the Supplementary materials, we have now added Table A in which patients characteristics are compared by CKD-EPI stage at the last visit before COVID-19 infection. This table shows that the majority of patients included in the study were in stages 3b (39.1%), 4 (28.4%) and 3a (21.5%).

The proportion of young patients (<50 yrs) in the PIRP cohort of AUSL Romagna is 4.17%, a low figure which stems from the inclusion criteria.

By including CKD stage in the regression of in-hospital mortality we showed that CKD stage 4 patients are at a higher risk of mortality. The survival curves and log-rank test provided in Supplementary materials already showed that mortality risk increased with increasing CKD stage.

8. Were the admissions to hospital for COVID-19, or where these unrelated admission with incidental tests? What was the proportion? Admitted patients will have had multiple tests, so there may be a detection bias here.

All patients that we reported as COVID+ because they were hospitalized had the COVID-19 diagnosis identified from a specific code in the hospital discharge records. We are not able to distinguish who was admitted already COVID+ from those who were infected during their hospital stay. 

9. The 95% hospitalisation rate seems extremely high – There must presumably be a major bias here towards the hospitalised patients as there was no routine testing in the community of the PRIP cohort so asymptomatic cases and mild disease that didn’t present for a test won’t have been captured. The COVID+ patients who were hospitalised will have had a test, but the COVID+ patients in the community may not. Unless there was routine testing, I don’t think these data have captured the full extent of COVID19+ in this population so I don’t think the 95% hospitalisation rate is valid.

We understand that our data refer to the first wave of the pandemic, in which no routine tests were performed and only symptomatic patients were seen in the health care facilities. In particular, at that time, CKD patients with COVID-19 were very likely to be hospitalized because of their complex clinical conditions. However, as we explained in a previous reply, most of these patients are in a particularly vulnerable state, and all of them have a direct connection with hospital nephrologists, as they are followed-up in a prevention program. We then hypothesize that the proportion of not tested COVID-19+ patients among them should be quite low.

Minor:

Line 183 the CFR is different to the rate at the start of the paragraph?

The CFR that was reported at line 183 referred to a sub-analysis that we made with the aim to compare the CFR that we found in PIRP patients to that found in an Italian survey on patients on dialysis. To obtain comparable figures, we restricted the CFR computation to a smaller time frame, as specified in lines 180-181.

CKD stages mentioned in the statistical analysis section of methods but not in the paper

We have added CKD stages as requested

I appreciate there are geographical and cultural differences in practice, but Im not sure its reasonable to call these “pre-dialysis” patients - they are very sick patients who would not be offered RRT in many healthcare settings – these patients would be on conservative care pathways typically.

We have used this terminology to reflect our selection of CKD stage 1 to 5 patients. As we explained in the ‘Study population’ section, our patients participate in a prevention program whose specific aim is to delay their initiation of RRT. Thus, our cohort includes both very sick patients who are in the course of being prepared for RRT and patients with slower progression who are maintained in conservative care. To avoid the possibility of confusion we have now changed the definition in the title from ‘pre-dialysis’ to ‘non-dialysis’ patients 

Figure 2a

I don’t think this is useful information and its very confusing to read – why is the date of onset of disease relevant? Why resulting in different death rates? Why are these seemingly arbitrary dates chosen?

The reason why we included the date of onset is that, in the initial phase of the pandemic, hospitals and healthcare systems were not prepared to treat the number of infected patients that showed up. Thus, patients with similar characteristics might have experienced different mortality because they were infected in the early phase rather in subsequent phases when healthcare system efficacy was higher. The timeframes were chosen according to the phases of the epidemic curve of COVID-19 in Italy, with shorter intervals during the high epidemic phase up to the beginning of May (when the strictest containment measures were enforced) and longer intervals afterwards. We have now specified in the ‘Statistical analysis’ section the rationale behind this.

---

## [Decision Letter · Decision Letter 1]

29 Jun 2021

COVID-19 incidence and mortality in non-dialysis chronic kidney disease patients

PONE-D-21-06894R1

Dear Dr. Gibertoni,

We’re pleased to inform you that your manuscript has been judged scientifically suitable for publication and will be formally accepted for publication once it meets all outstanding technical requirements.

**The revised version of the manuscript has definitely improved. The author have now adequately addressed all concerns raised by the reviewers.**

Kind regards,

Giuseppe Remuzzi

Academic Editor

PLOS ONE

Additional Editor Comments (optional):

Reviewers' comments:

Reviewer's Responses to Questions

**Comments to the Author**

1. If the authors have adequately addressed your comments raised in a previous round of review and you feel that this manuscript is now acceptable for publication, you may indicate that here to bypass the “Comments to the Author” section, enter your conflict of interest statement in the “Confidential to Editor” section, and submit your "Accept" recommendation.

Reviewer #1: All comments have been addressed

Reviewer #2: All comments have been addressed

2. Is the manuscript technically sound, and do the data support the conclusions?

Reviewer #1: (No Response)

Reviewer #2: Yes

3. Has the statistical analysis been performed appropriately and rigorously? 

Reviewer #1: (No Response)

Reviewer #2: Yes

4. Have the authors made all data underlying the findings in their manuscript fully available?

Reviewer #1: (No Response)

Reviewer #2: No

5. Is the manuscript presented in an intelligible fashion and written in standard English?

Reviewer #1: (No Response)

Reviewer #2: Yes

6. Review Comments to the Author

Reviewer #1: (No Response)

Reviewer #2: Previous comments have been addressed as much as possible, and limitations have been identified where appropriate.

7. PLOS authors have the option to publish the peer review history of their article (what does this mean?). If published, this will include your full peer review and any attached files.

Reviewer #1: **Yes: **Luuk Hilbrands

Reviewer #2: **Yes: **Eoin Daniel O'Sullivan

---

## [Editor Report · Acceptance letter]

1 Jul 2021

PONE-D-21-06894R1 

COVID-19 incidence and mortality in non-dialysis chronic kidney disease patients 

Dear Dr. Gibertoni:

I'm pleased to inform you that your manuscript has been deemed suitable for publication in PLOS ONE. Congratulations! Your manuscript is now with our production department. 

Kind regards, 

on behalf of

Prof. Giuseppe Remuzzi 

Academic Editor

PLOS ONE